# Longitudinal Correlations between Molecular Compositions of Stratum Corneum and Breast Milk Factors during Infancy: A Prospective Birth Cohort Study

**DOI:** 10.3390/nu16121897

**Published:** 2024-06-16

**Authors:** Risa Fukuda, Kyongsun Pak, Megumi Kiuchi, Naoko Hirata, Naoko Mochimaru, Ryo Tanaka, Mari Mitsui, Yukihiro Ohya, Kazue Yoshida

**Affiliations:** 1Division of Dermatology, National Center for Child Health and Development, Tokyo 157-8535, Japan; fukuda-rs@ncchd.go.jp (R.F.); tanaka-r@ncchd.go.jp (R.T.); 2Division of Biostatistics, Department of Data Management, Center of Clinical Research and Development, National Center for Child Health and Development, Tokyo 157-8535, Japan; 3Division of Research and Development, Pigeon Corporation, Ibaraki 300-2495, Japan; 4Center for Maternal-Fetal, Neonatal and Reproductive Medicine, National Center for Child Health and Development, Tokyo 157-8535, Japan; 5Allergy Center, National Center for Child Health and Development, Tokyo 157-8535, Japan

**Keywords:** ceramide, cholesterol, IgA, lactoferrin, natural moisturizing factor, stratum corneum hydration, TGF-β1, TGF-β2, transepidermal water loss, water content

## Abstract

Breast milk contains numerous factors that are involved in the maturation of the immune system and development of the gut microbiota in infants. These factors include transforming growth factor-β1 and 2, immunoglobin A, and lactoferrin. Breast milk factors may also affect epidermal differentiation and the stratum corneum (SC) barrier in infants, but no studies examining these associations over time during infancy have been reported. In this single-center exploratory study, we measured the molecular components of the SC using confocal Raman spectroscopy at 0, 1, 2, 6, and 12 months of age in 39 infants born at our hospital. Breast milk factor concentrations from their mothers’ breast milk were determined. Correlation coefficients for the two datasets were estimated for each molecular component of the SC and breast milk factor at each age and SC depth. The results showed that breast milk factors and molecular components of the SC during infancy were partly correlated with infant age in months and SC depth, suggesting that breast milk factors influence the maturation of the SC components. These findings may improve understanding of the pathogenesis of skin diseases associated with skin barrier abnormalities.

## 1. Introduction

Breast milk has evolved to serve as an interface between the maternal and infant immune systems, nourishing the infant and protecting it from disease until its own immune system matures. Breast milk is a species-specific food that is designed to meet the biological and psychological needs of newborns and infants. The composition of breast milk changes over time as the infant adapts from a wet to a dry environment [1].

Breast milk contains numerous factors that contribute to maturation of the immune system and development of the organs and gut microbiota in infants. These factors include transforming growth factors (TGFs)-β1 and 2, immunoglobin A (IgA), and lactoferrin [2,3]. Colostrum is secreted by the mammary glands for several days immediately after parturition and contains much higher levels of breast milk factors than mature milk [4]. Breast feeding is associated with low infant morbidity of gastrointestinal infections; inflammatory, respiratory, and allergic diseases; diabetes; and obesity [5,6,7,8,9,10]. 

The breast milk factor, TGF-β, plays a role in cell growth and development. The concentration of TGF-β1 but not TGF-β2 in breast milk during the first month after birth may be associated with eczema later in life [6]. IgA constitutes the majority of immunoglobulins present in breast milk; IgA positively correlates with TGF-β1 and 2 levels in colostrum [11,12] and plays an important role in the maturation of the infant’s intestinal immune system. As neonates produce little IgA on their own, IgA concentrations in breast milk are highest when colostrum is being produced; this also influences the composition of the infant’s gut microbiome, providing protection against milk allergy in infancy and colitis in adulthood [13,14]. The other breast milk factor, lactoferrin, is present in breast milk as well as in secretions and the vernix that covers the skin surface of newborns. Lactoferrin exerts a variety of host defense effects [15,16].

Breast milk contains a diverse bacterial population, and bacteria in mother’s milk can seed the infant’s gut and contribute to the development of the infant’s gut microbiota. Breastfed infants have a more stable and less diverse gut microbiota than formula-fed infants [17], and their gut microbiota profile, compared to that of non-breastfed infants, is affected even into adulthood [18]. Furthermore, the expressions of genes that regulate intestinal cell proliferation, differentiation, and barrier function in the fecal matter of breastfed and formula-fed infants are different [19].

Recent studies have shown that these breast milk factors influence epidermal differentiation and barrier function [1,20,21]. The molecular components of the stratum corneum (SC)—including water content, intercellular lipids such as ceramide and cholesterol, and natural moisturizing factor (NMF) and its component, lactic acid—are important for maintaining epidermal barrier function. After birth, dynamic changes in water content and NMF levels in the epidermis occur to adapt to the dry environment [22]. Bovine colostrum promotes keratinocyte division, motility, and cell differentiation [23,24]; induces the expression of keratinocyte differentiation markers (including keratin 1, involucrum, and filaggrin); and increases the levels of NMF and lactic acid in the epidermis by inducing the synthesis of caspase 14 and bleomycin hydrolase, which are involved in filaggrin maturation [24].

This study aimed to examine the longitudinal changes and correlations between the molecular composition of the SC of infants and breast milk factors of their mother’s milk over the course of the first year of infancy and to investigate the involvement of breast milk factors in the maturation of SC components.

## 2. Materials and Methods

### 2.1. Participants

This was a prospective cohort study that included mothers and their infants who were delivered at the National Center for Child Health and Development between July 2019 and October 2020. For patient recruitment, posters and brochures about the study were displayed and distributed at the dermatology and obstetrics and gynecology outpatient clinics, and 63 pregnant women attending our hospital inquired about the study. After their deliveries, 19 of these women were excluded prior to enrollment, and 44 infants who met the inclusion criteria of <7 days of age and their mothers were enrolled in the study. Subsequently, five infants and their mothers were excluded and, finally, 39 infants and their mothers underwent the measurements. None of the study participants met the following exclusion criteria: infants whose parents did not understand Japanese, infants with known serious complications, or infants deemed unsuitable for participation by the dermatologist. The participants’ family members provided written informed consent for the publication of their details. This study was approved by the Institutional Ethics Committees of the National Center of Child Health and Development (approval number: 2235).

### 2.2. Measurement of the Molecular Composition of the Infants’ SC

Water content; levels of NMF, ceramide, cholesterol, and lactic acid; and SC thickness were estimated using a confocal Raman spectrometer (CRS; Model 3510 Skin Composition Analyzer; RiverD B.V., Rotterdam, The Netherlands). Transepidermal water loss (TEWL) and SC hydration were measured using a Tewameter (TM300; Courage + Khazaka electronic GmbH, Köln, Germany) and Corneometer (CM 825; Courage + Khazaka electronic GmbH), as previously reported [25]. Measurements were performed at 0 (1–7 days old), 1 (±14 days), 2 (+1 month), 6 (5–10 months), and 12 (11–24 months) months (Figure 1). Measurements were taken on the skin around the middle of the calf at distances of 4 μm in an environment maintained at 22 °C ± 2 °C and 50% ± 10% humidity. The application of moisturizers at the measurement sites was prohibited on the day of measurement.

### 2.3. Measurement of Mothers’ Breast Milk

The mothers completed questionnaires that included their background information, mode of delivery, family history of allergies (atopic dermatitis (AD), asthma, allergic rhinitis, allergic conjunctivitis, hay fever, or food allergies), and skin care routine used for their infants. The mothers provided breast milk samples when their infants were 0, 1, 2, and 6 months of age, not 12 months of age (Figure 1).

Breast milk factors, including TGF-β1 and 2, IgA, and lactoferrin, were measured using enzyme-linked immunosorbent assay, as was previously reported [4]. Breast milk collected from the first to seventh day postpartum was defined as colostrum.

### 2.4. Statistical Analysis

To investigate the association between maternal and infant characteristics, frequencies and percentages were calculated. For the molecular compositions of the SC measured using CRS, we drew the mean ± standard deviation (SD) per SC depth, and median area under the receiver operating characteristic curves (AUCs) and interquartile ranges (IQRs) were calculated for depths of 0 to 4, 4 to 8, 8 to 20, and 0 to 20 μm. Median values of TEWL and SC hydration, as well as TGF-β1 and 2, IgA, and lactoferrin, were shown for each month of age. Spearman’s correlation coefficients were estimated for each molecular component in the SC (including water content; NMF, ceramide, cholesterol, and lactic acid levels; TEWL; SC hydration; and SC thickness) and breast milk factors (including TGF-β1 and 2, IgA, and lactoferrin levels) at each age (0, 1, 2, 6, and 12 months) and SC depth. Correlation coefficients of 0.1 to 0.3 were considered to represent a weak correlation, 0.4 to 0.6 a moderate correlation, and >0.6 a strong correlation. Since this study was not conducted using a sample size calculation, the *p*-value was used as a reference index. All statistical analyses were performed using R version 4.1.2. This study was exploratory; therefore, the multiplicity of the tests was not considered.

## 3. Results

### 3.1. Patients’ Demographics

Questionnaires were administered to 39 mothers at 0 months, 35 at 1 month, 32 at 2 and 6 months and 33 at 12 months. In Table 1, more than 90% of the participating infants had a family history of allergies (Table 1). Less than 15% of the infants were exclusively breastfed during the entire period, with most being mixed nutrition (Table 2). No significant differences in the sex or skin care regimens of infants were observed (Table 2).

### 3.2. Longitudinal Molecular Changes in the Infants’ SC

Using CRS, the SC of 39, 35, 32, 32, and 33 infants were measured at 0 months, 1 month, 2 and 6 months, and 12 months of age, respectively. The ages were as follows: 0 months, 4.0 (3.0–5.0) days; 1 month, 30.0 (27.0–33.0) days; 2 months, 65.0 (62.0–70.2) days; 6 months, 186.0 (182.0–232.5) days; and 12 months, 366.8 (362.0–374.0) days. Figure 1 presents the mean ± SD of water content in the SC, as well as the NMF, ceramide, cholesterol, and lactic acid concentrations in the SC, for each age group, calculated using the CRS results. Appendix A shows the median AUC for each SC depth in each group. The median and IQR for AUC by depth for each age group are presented in Appendix A.

The SC thickness estimated using CRS was the thickest (22.485 (21.093–25.440) μm) at 0 months and gradually thinned to 14.322 (13.571–16.964 mm) at 2 months (Appendix A). Thereafter, the SC thickness gradually increased with time, without reaching the value measured at 0 months (Appendix A). The water content was lowest (0–20 μm, 754.496 (690.043–822.310) mass-%·μm) at 0 months but increased and reached its highest levels at 2 months (0–20 μm, 1012.503 (980.740–1040.574) mass-%·μm). At all ages, the water content was lowest in the upper layers of the SC and increased with increasing depth (Figure 2a and Appendix A; Appendix A). The highest levels of NMF were found in the middle layer of the SC, with the highest level (0–20 μm, 11.867 (9.572–15.766) a.u.·μm) measured at 0 months and the lowest level (0–20 μm, 5.844 (4.615–9.088) a.u.·μm) measured at 2 months. Thereafter, NMF levels gradually increased with age (Figure 2b and Appendix A; Appendix A). Both the ceramide and cholesterol levels were higher in the upper SC and decreased with increasing SC depth. These levels were highest at 0 months (ceramide: 0–20 μm, 2687.160 (2448.113–3017.410) a.u.·μm; cholesterol: 0–20 μm, 1.291 (1.158–1.489) a.u.·μm) and decreased with age (Figure 2c,d and Appendix A; Appendix A). The levels of lactic acid were enriched in the upper SC and were highest at 6 months (0–20 μm, 537.829 (454.906–722.808) a.u.·μm; Figure 2e and Appendix A; Appendix A).

SC hydration was the lowest at 0 months (26.20 (20.25–30.45) a.u.), increased with age, peaked at 6 months (56.25 (47.50–68.05) a.u.), and then was slightly decreased at 12 months (Appendix A; Appendix A). This change correlated with the water content measured using CRS. The TEWL levels were low at 0–1 months, increased slightly from 2 to 6 months, and showed a decreasing tendency at 12 months (Appendix A; Appendix A).

### 3.3. Longitudinal Changes of Breast Milk Factors

The median values and IQRs for TGF-β1 and 2, IgA, and lactoferrin at each month are shown in Figure 3 and Appendix A. Breast milk data were obtained from 39 mothers at 0 months, 35 mothers at 1 month, and 32 mothers at 2 and 6 months. The concentrations of all breast milk factors were highest in colostrum and decreased as breast milk matured. In particular, the levels of TGF-β1 and 2 and IgA showed marked decreases from 0 to 1 months (Figure 3; Appendix A).

### 3.4. Spearman’s Correlation Analysis of the Molecular Components in the SC and Breast Milk Factors

The Spearman’s correlation analysis of the molecular components of the SC and breast milk factors is shown in Figure 4. Water content showed a tendency toward a weak to moderate negative correlation at all ages with all breast milk factors (Figure 4a). TGF-β1 and 2 at 0 months showed a weak to moderate negative correlation in the upper layer at 0 and 1 months. TGF-β2 levels at 6 months also showed a weak to moderate negative correlation with water content in all layers at 12 months. IgA levels at 0 months showed a weak to moderate negative correlation in all layers in the same month (Figure 4a).

NMF levels showed a tendency toward a weak to moderate positive correlation with all breast milk factors throughout the study period (Figure 4b). However, TGF-β1 was uncorrelated with NMF for the most part throughout the study period. The TGF-β2 level at 1 and 6 months had a weak to moderate positive correlation with the NMF level in the shallow layer during the same months of age. The lactoferrin levels at 2 months showed a strong positive correlation in the deep layer at 12 months (Figure 4b). Levels of both ceramide and cholesterol had an overall weak correlation with all breast milk factors throughout the entire period (Figure 4c,d). Regarding ceramide, TGF-β2 at 0 months showed a moderate positive correlation with ceramide at 6 months in the upper layers (Figure 4c). Regarding cholesterol, TGF-β1, IgA, and lactoferrin at 0 months showed a weak positive correlation with cholesterol after 2 months. TGF-β1 and 2 after 1 month showed a weak negative correlation in subsequent months (Figure 4d). Lactic acid levels showed a positive correlation with all breast milk factors in all the layers. In particular, the IgA levels at 2 months showed a strong positive correlation with the lactic acid levels in the deep layer at 12 months (Figure 4e).

Spearman’s correlation analysis of the median SC hydration and TEWL with breast milk factors was performed for each age group (Figure 5 and Appendix A). Both TGF-β1 and 2 levels at 0–2 months showed a weak to strong positive correlation with SC hydration at 2 months. IgA and lactoferrin at 0 and 1 months showed a weak to moderate negative correlation with SC hydration during the intermediate months. IgA at 0 and 1 months showed a moderate negative correlation with TEWL at 1 month. Lactoferrin at 2 months showed a moderate positive correlation with TEWL at 12 months (Figure 5 and Appendix A). Over the study period, SC thickness showed a weak to moderate positive correlation with all breast milk factors (Appendix A). In particular, TGF-β1 at 2 months showed a strong positive correlation with SC thickness at 6 months (Appendix A).

## 4. Discussion

In this study, we examined the correlations between the molecular components of the SC and breast milk factors during infancy. This study revealed that these correlations were partly dependent on age (in months) and SC depth. These results suggest that breast milk factors play important roles in the maturation of the infant skin barrier.

Regarding patient characteristics, less than 15% of the infants were exclusively breastfed during the entire period, with most being mixed nutrition (Table 2). In this study, the responses to the questionnaire showed that milk production was adequate, and the sampling volumes of breast milk were sufficient, suggesting that the amounts of breast milk were adequate. Furthermore, most Japanese mothers who choose mixed nutrition supplement formula milk after breast milk feeding. Therefore, we assume that breast milk intake was sufficient even with mixed nutrition.

The SC thickness is thinnest at approximately 1 to 6 months of age [25,26]. The SC then thickens until approximately 10 years of age, after which it becomes as thick as that of an adult. In the present study, as in previous reports, the SC thickness was thinnest at 2 months of age and then became thicker [27]. The measured values and degree of change in water content and levels of NMF, ceramide, cholesterol, and lactic acid were also almost consistent with those reported previously [22,25]. Molecular changes in the SC may have been more accurately identified in the present study because the development in the same individuals was followed over the study period.

The breast milk factors TGF-β1 and 2, IgA, and lactoferrin are involved in protecting the epidermal barrier [1,20,21]. Bovine colostrum promotes keratinocyte division, motility, and cell differentiation; induces the expression of epidermal differentiation markers (including keratin 1, involucrin, and filaggrin); and leads to the synthesis of caspase 14 and bleomycin hydrolase, thereby increasing NMF and lactate levels [23,24]. Oral administration of lactoferrin derived from bovine milk may inhibit increased TEWL, decreased water content, abnormal epidermal hyperplasia, and cell apoptosis by reducing ultraviolet-stimulated interleukin (IL)-1β levels. Furthermore, orally administered lactoferrin also acts on lipid metabolism, decreasing the synthesis and absorption of cholesterol in the serum, increasing cholesterol excretion, and decreasing sebum (especially triacylglycerol) secretion in the SC [28,29,30,31,32]. Human breast milk is also involved in the development of lipid synthesis pathways in early life by promoting prostaglandin synthesis in skin fibroblasts [33]. On the other hand, IgA in breast milk is involved in the maturation of the gut microbiota during infancy [34]. Although it remains unclear whether there is a direct association between IgA and SC components in infants, nutritional methods and modes of delivery affect the diversity of the gut microbiota [35,36,37,38,39,40], and gut microbiota, in turn, affect skin maturation [41]. Taken together, breast milk factors act on the SC components, gut microbiota, and immune function and consequently may regulate the maturation of the skin barrier function.

The weak correlation of water content in this study may be due to the absence of breast milk factors that act directly on an increase in the water content level. However, the water content is presumed to be retained because of elevated NMF, lactic acid, and lipid levels. Furthermore, because lactic acid is also derived from sweat [42], water content is influenced by seasonal skin conditions [43]. Differences were noted in the correlations between water content measured using CRS and SC hydration measured using a Corneometer for each breast milk factor. Contrary to correlations for SC hydration, negative correlations with each breast milk factor at all months were observed for water content. These differences may be related to the negative correlation observed between water content and SC hydration values [44] and the limited depth to which SC hydration can be measured using the Corneometer. CRS is a noninvasive and highly sensitive optical method for measuring SC components at depth. Therefore, CRS is more suitable for measuring SC components in infants, as they are more sensitive than adults [25].

The positive correlation of NMF with all breast milk factors is supported by filaggrin expression and synthesis of filaggrin-degrading enzymes being induced in bovine colostrum [23,24]. Both ceramide and cholesterol showed an overall weak correlation with all breast milk factors over the entire period, but both TGF-β1 and 2 at 0 months, especially TGF-β2, were positively correlated with SC lipids in subsequent months. These results might be related to the finding that levels of TGF-β by the age of 1 month act to prevent future eczema development [6]. Furthermore, the levels of SC lipids may also be affected by lactoferrin and vernix. Although lactoferrin and SC lipids tended to be positively correlated in this study, orally administered lactoferrin affects lipid metabolism, decreases cholesterol synthesis and absorption, increases cholesterol excretion, and may decrease sebum levels and lipid secretion in the SC [29,31,32]. It is also suggested that vernix containing cholesterol and ceramide results in higher levels of ceramide and cholesterol in the SC at 0 months [45].

TEWL showed partial weak to moderate positive correlations with all breast milk factors throughout the study period. The participating infants were considered high-risk infants for developing allergies due to their high family history of allergies, which may have affected the molecular components in the SC, including causing an increase in TEWL [46]. Studies on the associations between the molecular components of the SC and breast milk factors in infants with and without AD will be considered in the future.

A limitation of this study was that it was an exploratory study, wherein the sample size was defined by the feasibility of the study. In addition, factors that affected the values of the molecular components of the large proportion of infants with a family history of allergies among the participating infants, and season at the time of measurement, other nutrients and bioactive substances in breast milk, such as lactalbumin, lactoferrin, lactose, and lipids, and the potential influence of the circadian clock on TGF-β in breast milk [47] were not considered. The frequency of bathing and types and use frequency of soaps and moisturizers were not included in the analysis, but almost all participants used these products, suggesting an influence of these on the SC components. Although breast milk factors have a variety of potential effects on infants, few studies have examined the correlation between breast milk factors and molecular changes in the infant SC. This study is the first to examine these changes and correlations over a 1-year period, and the results may provide important clues as to how breast milk factors may influence SC maturation during infancy and beyond.

## 5. Conclusions

This is the first study to investigate the longitudinal correlation between the molecular components of the SC and breast milk factors during infancy. A partial correlation was found between SC components and breast milk factors over time, suggesting that breast milk factors may influence SC maturation during infancy. A full understanding of these changes may aid in elucidating the pathogenesis of skin diseases associated with skin barrier abnormalities, including AD, and may help in determining the pathogenesis of these diseases and developing targeted topical therapies.

## Figures and Tables

**Figure 1 nutrients-16-01897-f001:**
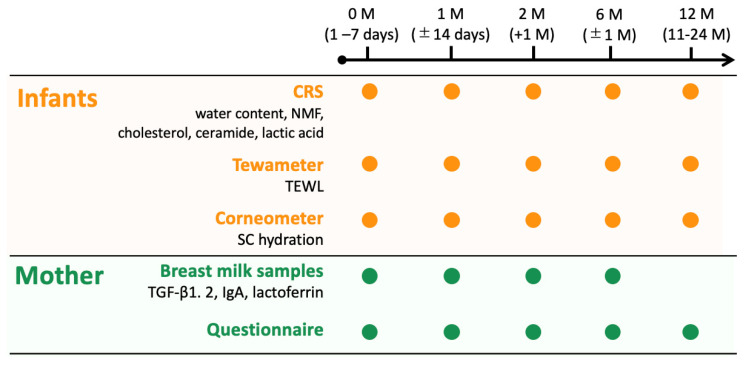
Flowchart outlining the methodological procedures for following up the mother–infant pairs. The stratum corneum (SC) components of the infants were measured using a confocal Raman spectrometer (CRS), Tewameter, and Corneometer at 0, 1, 2, 6, and 12 months. The mothers completed questionnaires and provided breast milk samples when their infants were 0, 1, 2, and 6 months.

**Figure 2 nutrients-16-01897-f002:**
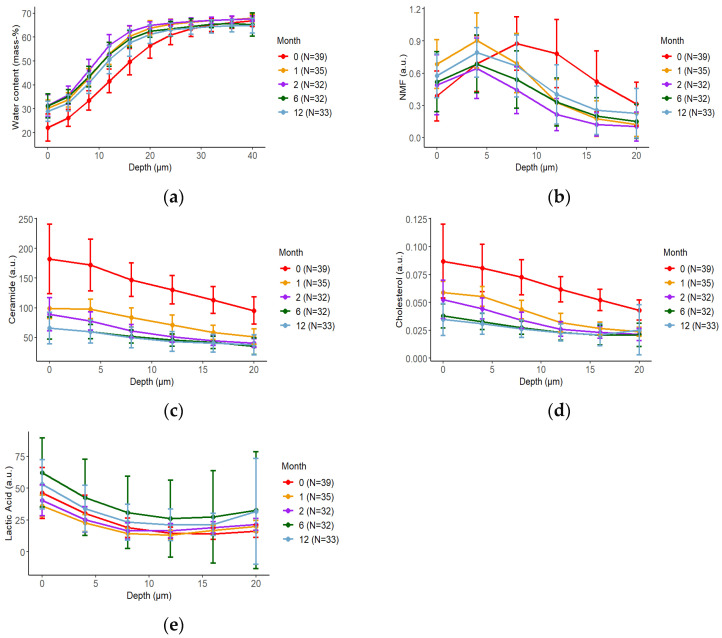
Changes in the molecular composition of the stratum corneum (SC) over time based on confocal Raman spectroscopy (CRS) measurements. The depth profile (mean ± standard deviation) of water content (**a**), natural moisturizing factor (NMF) (**b**), ceramide (**c**), cholesterol (**d**), and lactic acid (**e**) for each month of age are shown. The data are marked at 0 months (red), 1 month (orange), 2 months (purple), 6 months (green), and 12 months (light blue). Error bars indicate 95% confidence intervals. The mean of five measurements was considered as each participant’s representative value.

**Figure 3 nutrients-16-01897-f003:**
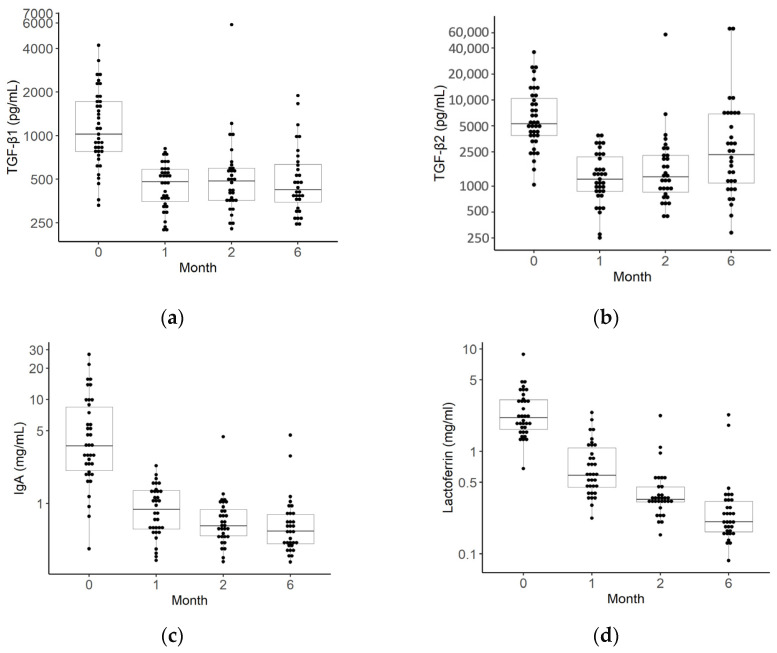
Changes over time in transforming growth factors (TGFs)-β1 and 2, immunoglobin A (IgA), and lactoferrin levels in breast milk. The median TGF-β1 (**a**), TGF-β2 (**b**), IgA (**c**), and lactoferrin (**d**) levels in the breast milk of mothers. The black dots indicate the value of the breast milk factors for each month of age (39 for 0 months, 35 for 1 month, 32 for 2 months, 32 for 6 months, and 33 for 12 months).The box plots represent medians with 25th and 75th percentile values, min–max range, and outliers.

**Figure 4 nutrients-16-01897-f004:**
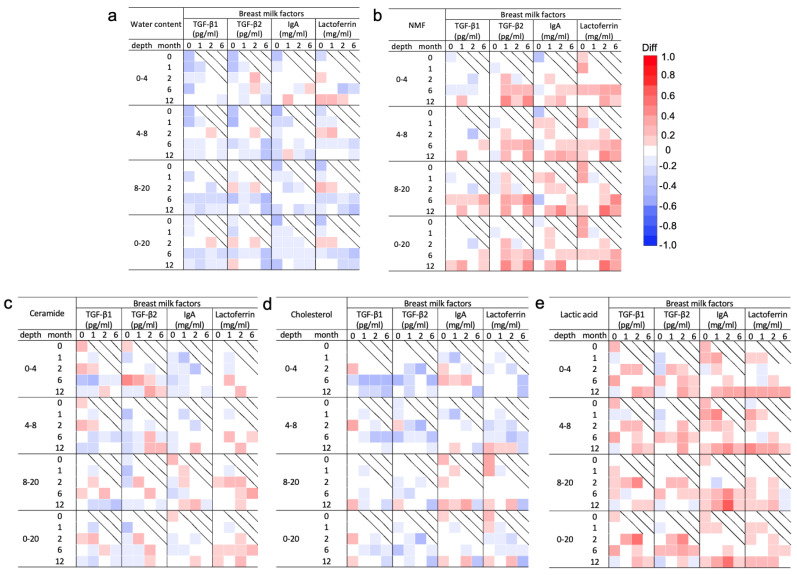
Correlation analysis of molecular components in the stratum corneum (SC) measured by confocal Raman spectroscopy and breast milk factors for each month of age. Correlation coefficients and 95% confidence intervals between the median area under the receiver operating curve by depth (0–4, 4–8, 8–20, and 0–20 μm) for water content (**a**), natural moisturizing factor (NMF) (**b**), ceramide (**c**), cholesterol (**d**), and lactic acid (**e**) measured at 0, 1, 2, 6, and 12 months and levels of breast milk factors including transforming growth factor (TGF)-β1 and 2, immunoglobin A (IgA), and lactoferrin measured at 0, 1, 2, and 6 months are shown (Appendix A). Water content and ceramide and cholesterol levels tended to be negatively correlated with all breast milk factors during the entire period, while NMF and lactoferrin levels were positively correlated with all breast milk factors.

**Figure 5 nutrients-16-01897-f005:**
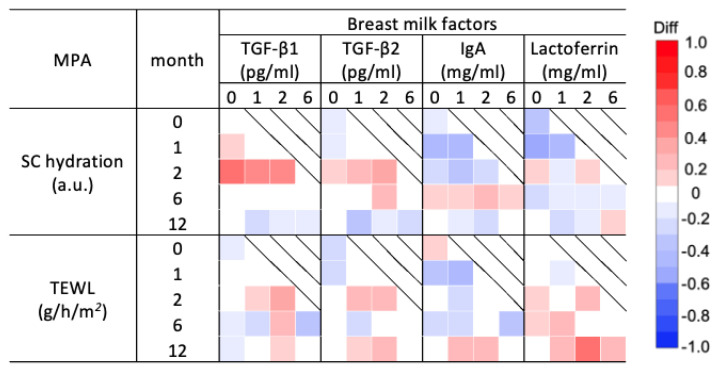
Correlation analysis of the stratum corneum (SC) hydration and transepidermal water loss (TEWL) in the SC and breast milk factors for each month of age. Correlation coefficients and 95% confidence intervals between the median of SC hydration and TEWL measured at 0, 1, 2, 6, and 12 months and the levels of breast milk factors including transforming growth factors (TGFs)-β1 and 2, immunoglobin A (IgA), and lactoferrin measured at 0, 1, 2, and 6 months are shown (Appendix A). TGF-β1 and 2 levels correlated positively with SC hydration for almost the entire period, while IgA and lactoferrin levels correlated negatively. TEWL was partially positively correlated with all breast milk factors.

**Table 1 nutrients-16-01897-t001:** Basic characteristics of the participants.

Factors	Classes	Total (*N* = 39) *N* (%)
Infants’ sex	Male	21 (53.8%)
	Female	18 (46.2%)
Birth weight	<2500 g	0 (0.0%)
	2500−4000 g	39 (100.0%)
	≥4000 g	0 (0.0%)
Family history of allergies	Yes	37 (94.9%)
	No	2 (5.1%)
Mother’s smoking status	Yes	9 (23.1%)
	No	30 (76.9%)
Parity	Primipara	29 (74.4%)
	Multipara	10 (25.6%)
Gestational age	<37 weeks	0 (0.0%)
	37 to 41 weeks	39 (100.0%)
	≥42 weeks	0 (0.0%)
Age at delivery	20 s	4 (10.3%)
	30 s	21 (53.8%)
	40 s	14 (35.9%)
Delivery mode	Vaginal birth	26 (66.7%)
	Cesarean section	13 (33.3%)
Colostrum secretion	Before birth	7 (17.9%)
	After birth	32 (82.1%)
Colostrum collection day	≤3 days	15 (38.5%)
	4 days	9 (23.1%)
	≥5 days	15 (38.5%)

**Table 2 nutrients-16-01897-t002:** Variable characteristics of the participants.

Factors	Classes	Months				
		0	1	2	6	12
		Total (*N* = 39) *N* (%)	Total (*N* = 35) *N* (%)	Total (*N* = 32) *N* (%)	Total (*N* = 32) *N* (%)	Total (*N*= 33) *N* (%)
Nutrition source	Breast	3 (7.7%)	5 (14.3%)	4 (12.5%)	5 (15.6%)	4 (12.1%)
	Formula	1 (2.6%)	0 (0.0%)	0 (0.0%)	0 (0.0%)	1 (3.0%)
	Mix	35 (89.7%)	30 (85.7%)	28 (87.5%)	23 (71.9%)	11 (33.3%)
Bath frequency	No bath	28 (71.8%)	0 (0.0%)	0 (0.0%)	0 (0.0%)	0 (0.0%)
	Every few days	0 (0.0%)	0 (0.0%)	0 (0.0%)	0 (0.0%)	0 (0.0%)
	Once daily	7 (17.9%)	34 (97.1%)	30 (93.8%)	26 (81.2%)	28 (84.8%)
	Twice or more daily	0 (0.0%)	1 (2.9%)	2 (6.2%)	6 (18.8%)	4 (12.1%)
	Frequency unknown	4 (10.3%)	0 (0.0%)	0 (0.0%)	0 (0.0%)	1 (3.0%)
Wash with soap in bath	Yes	6 (15.4%)	34 (97.1%)	32 (100.0%)	32 (100.0%)	33 (100.0%)
	No	33 (84.6%)	1 (2.9%)	0 (0.0%)	0 (0.0%)	0 (0.0%)
Moisturizer use (face)	No use	8 (20.5%)	0 (0.0%)	0 (0.0%)	0 (0.0%)	0 (0.0%)
	Less than once daily	3 (7.7%)	0 (0.0%)	0 (0.0%)	1 (3.1%)	1 (3.0%)
	Once daily	13 (33.3%)	23 (65.7%)	16 (50.0%)	11 (34.4%)	12 (36.4%)
	Twice or more daily	6 (15.4%)	11 (31.4%)	16 (50.0%)	19 (59.4%)	17 (51.5%)
	Frequency unknown	9 (23.1%)	1 (2.9%)	0 (0.0%)	1 (3.1%)	3 (9.1%)
Moisturizer use (legs)	No use	15 (38.5%)	1 (2.9%)	0 (0.0%)	0 (0.0%)	0 (0.0%)
	Less than once daily	1 (2.6%)	0 (0.0%)	0 (0.0%)	0 (0.0%)	1 (3.0%)
	Once daily	10 (25.6%)	27 (77.1%)	17 (53.1%)	11 (34.4%)	14 (42.4%)
	Twice or more daily	6 (15.4%)	6 (17.1%)	13 (40.6%)	15 (46.9%)	15 (45.5%)
	Frequency unknown	7 (17.9%)	1 (2.9%)	2 (6.2%)	6 (18.8%)	3 (9.1%)
Breast or nipple problem	Yes	27 (69.2%)	16 (45.7%)	7 (21.9%)	7 (21.9%)	1 (3.0%)
	No	12 (30.8%)	19 (54.3%)	25 (78.1%)	25 (78.1%)	31 (93.9%)
Nipple care	Yes	20 (51.3%)	19 (54.3%)	10 (31.2%)	5 (15.6%)	2 (6.1%)
	No	19 (48.7%)	16 (45.7%)	22 (68.8%)	27 (84.4%)	29 (87.9%)

## Data Availability

Data will be provided upon request.

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
