# Peer review of "Longitudinal Correlations between Molecular Compositions of Stratum Corneum and Breast Milk Factors during Infancy: A Prospective Birth Cohort Study"

_nutrients, 2024, doi:10.3390/nu16121897_

Round 1

Reviewer 1 Report

Comments and Suggestions for Authors

Study of Breast Milk Factors (TGF, Lactoferrin, IgA) and Their Correlation with the Composition of the Stratum Corneum.

Introduction: Seems accurate.

Materials and Methods: 39 mothers and 39 infants were selected for the study. The authors should include the exclusion criteria for the study. The presence of dermatitis or atopy in infants may influence the composition of the stratum corneum, so it would be interesting to include how their occurrence has been controlled and if this has modified the results obtained.

The authors combine a very broad period at the time of study of 12 months (between 11 and 24 months). This may alter the characteristics of the skin.

The authors include means and medians and use the Pearson correlation test. They should perform tests for normality (Shapiro test) and if the variables are not normally distributed, use the Spearman correlation test. Similarly, the presentation of results should be unified according to their normality as mean and SD or median and interquartile range.

Results: Less than 15% of the children studied received exclusive breastfeeding, with mostly mixed feeding. The authors should analyze how this may influence the results. They are studying the relationship between factors present in breast milk and the maturation of the stratum corneum. Not receiving exclusive breastfeeding may influence the values of the maternal factors studied and thus the results obtained. In the absence of a control group and the low proportion of exclusive breastfeeding, it may be doubted whether the results in the composition of the stratum corneum are related to the maternal factors studied and not just the result of a mathematical association.

The correlation tables are difficult to see and interpret. The authors should add in the footnotes of the tables how they should be interpreted.

There is no reference to how they evaluated the presence of allergy history (95% of the sample), the use of baths, soaps, or creams in the results obtained, and in the possible alteration of skin composition. The data are included but no assessment is made.

Discussion: The authors should include their assessments of the aspects mentioned above.

Conclusion: Suggesting that the factors studied in breast milk influence the maturation of the stratum corneum requires a study with a larger sample size where confounding variables such as the high rate of mixed breastfeeding, the use of soaps or creams, or bathing routines are controlled.

Reviewer 2 Report

Comments and Suggestions for Authors

The article titled " Longitudinal Correlations Between Molecular Compositions of Stratum Corneum and Breast Milk Factors During Infancy: A Prospective Birth Cohort Study " (nutrients-3015965) is submitted to the journal under the section of "Pediatric Nutrition."

The abstract encapsulates essential components of the work, providing a comprehensive preview of the content.

The introduction is well-structured, addressing current knowledge on the general benefits of breastfeeding and specifically on stratum corneum (SC) development. The bibliography cited is relevant, enabling a thorough understanding of the study.

However, it is suggested that the conclusion of the introduction, where the objective is presented, should explicitly state the study's aim.

Regarding the Materials and Methods section, it is important to specify that this is a prospective cohort design. Details should include how parents or guardians of infants were recruited, the inclusion and exclusion criteria applied, and ethical approval obtained from the committee overseeing the study. For instance, while the Results section indicates that 90% of participants have family history of allergies, additional details are needed.

It would be beneficial to incorporate a flowchart outlining the methodological procedures for following mother-infant pairs in the study.

Furthermore, calculating the necessary sample size for this study is crucial. Whether this is the first such study conducted with a convenience and exploratory sample, this aspect is fundamental as it conditions the assessment of statistical significance or insignificance in the findings.

Regarding the results, a comparison test among different groups to determine uniformity or disparity between options is recommended, along with presenting a 95% confidence interval for each proportion.

Table 1b should be identified separately as Table 2, distinct from Table 1a, which details baseline characteristics. Table 1b illustrates follow-up progression, necessitating inter-group comparisons.

Regarding Figures 1 and 2, if the p-value is 0.001503, rounding it to 0.002 is appropriate. These figures are highly informative and encapsulate the study's results.

Figure 3 should be simplified; intricate details might be moved to an appendix or supplementary material due to complexity, aiding in result integration.

Figure 4 is more comprehensible.

The Discussion should align closely with the stated objectives, integrating findings with existing literature and offering insightful commentary on stratum corneum formation and its relation to breastfeeding.

Among the limitations, it should be noted that the quality of breast milk was not assessed—an important aspect often exhibiting significant variations, which should be considered in future studies. Clarification on sample size and the predominantly allergic participant profile is warranted.

The conclusion aligns well with the presented results.

Round 2

Reviewer 1 Report

Comments and Suggestions for Authors

The authors have adequately addressed the questions raised and have made the necessary modifications to the text. The manuscript has been clearly improved.

Reviewer 2 Report

Comments and Suggestions for Authors

I have carefully reviewed the new version of the manuscript titled " Longitudinal Correlations Between Molecular Compositions of Stratum Corneum and Breast Milk Factors During Infancy: A Prospective Birth Cohort Study " (nutrients-3015965), as well as the authors' response to each of the suggestions made.

I consider that the authors have followed all the suggestions except those they have justified.

Overall, I consider that the article provides relevant information, being the first study to investigate the longitudinal correlation between the molecular components of the SC and breast milk factors during infancy. A partial correlation was found between SC components and breast milk factors over time, suggesting that breast milk factors may influence SC maturation during infancy.